# Amphiphilic Silver Nanoparticles for Inkjet-Printable Conductive Inks

**DOI:** 10.3390/nano12234252

**Published:** 2022-11-29

**Authors:** Irena Ivanišević, Marin Kovačić, Marko Zubak, Antonia Ressler, Sara Krivačić, Zvonimir Katančić, Iva Gudan Pavlović, Petar Kassal

**Affiliations:** 1Faculty of Chemical Engineering and Technology, University of Zagreb, Marulićev trg 19, 10000 Zagreb, Croatia; 2Faculty of Engineering and Natural Sciences, Tampere University, Korkeakoulunkatu 6, P.O. Box 589, 33014 Tampere, Finland

**Keywords:** silver nanoparticles, amphiphilic particles, density functional theory, conductive ink, inkjet printing, flexible electronics

## Abstract

The large-scale manufacturing of flexible electronics is nowadays based on inkjet printing technology using specially formulated conductive inks, but achieving adequate wetting of different surfaces remains a challenge. In this work, the development of a silver nanoparticle-based functional ink for printing on flexible paper and plastic substrates is demonstrated. Amphiphilic silver nanoparticles with narrow particle size distribution and good dispersibility were prepared via a two-step wet chemical synthesis procedure. First, silver nanoparticles capped with poly(acrylic acid) were prepared, followed by an amidation reaction with 3-morpholynopropylamine (MPA) to increase their lipophilicity. Density functional theory (DFT) calculations were performed to study the interactions between the particles and the dispersion medium in detail. The amphiphilic nanoparticles were dispersed in solvents of different polarity and their physicochemical and rheological properties were determined. A stable ink containing 10 wt% amphiphilic silver nanoparticles was formulated and inkjet-printed on different surfaces, followed by intense pulsed light (IPL) sintering. Low sheet resistances of 3.85 Ω sq^–1^, 0.57 Ω sq^–1^ and 19.7 Ω sq^–1^ were obtained for the paper, coated poly(ethylene terephthalate) (PET) and uncoated polyimide (PI) flexible substrates, respectively. Application of the nanoparticle ink for printed electronics was demonstrated via a simple flexible LED circuit.

## 1. Introduction

The key requirement for the mass production of printed electronic devices on lightweight and flexible substrates is a simple, low-cost, environmentally friendly and effective process such as inkjet printing [1,2]. Inkjet printing is a non-contact printing technology which can be implemented into the high-speed roll-to-roll manufacturing of printed electronics [3]. Today, inkjet printing technology is the most widely used technology for the additive manufacturing of conductive patterns on flexible substrates utilizing functional building blocks. The main field of interest in the application of inkjet printing technology is in the electronics industry, especially in the areas of wearables and smart packaging [4,5,6]. Among the various inkjet processes, the most suitable ones are based on the generation of droplets by electrochemically induced pressure waves. Hence, the drops are formed only when required, while the desired pattern geometry can be achieved by moving either the printer head or the substrate.

Among the conductive materials developed for inkjet printing, such as metallic/bimetallic nanoparticles and core–shell structures (i.e., gold, copper, nickel, etc.), as well as inorganic non-metallic (graphite, graphene and carbon nanotubes) and conductive polymer materials, silver nanoparticles (AgNPs) nowadays predominate as the best conductive materials for the fabrication of printed electronics [7,8]. The following conductive inks based on pristine [9] and bimetallic core–shell [10] nanosilver structures have been successfully used for printing: conductive tracks [11], Ag-Zn batteries [12], write-once-read-many (WORM) memories [13], differential temperature sensors [14], textile capacitors [15], printed antennas [16], electrodes for the fabrication of solar cells [17] and electrochemical (bio)sensors [18].

The quality of printing depends directly on the quality, type and amount of ink. To date, there is no universal ink formulation; inkjet inks are designed for use in a specific printing system or printhead [4]. During the printing process, the ink undergoes various stages: it is firstly produced in a carrier medium, stored in a glass container or printer cartridge, processed into droplets, adsorbed or absorbed and fixed by the substrate, subjected to post-printing processing and, finally, used in the form of a functional product. Despite numerous efforts by researchers, and studies conducted, the formulation of conductive ink remains one of the biggest challenges in inkjet printing technology [19]. Nowadays, major research and development efforts in academia and industry are directed towards the formulation of multipurpose inks, i.e., inks that can be printed on different (flexible) substrates.

In general, most functional materials, including dispersions of silver nanoparticles, are thermodynamically unstable species [20]. To prevent agglomeration of the particles in the dispersion and to allow good adhesion between the printed silver ink pattern and the desired surface, the presence of a stabilizing (capping) agent is of utmost importance [21]. The stabilizers most commonly reported in the literature for formulating stable AgNP inks are neutral polymers such as polyvinylpyrrolidone [22] and polyelectrolytes such as poly(acrylic acid) [23]. In addition, the choice of stabilizer determines the hydrophilic or hydrophobic properties of the particle, which consequently determine the dispersibility of the particles in solvents of different polarity and wetting of the surface. While hazardous organic solvents are avoided, aqueous inks have increased surface tension, which severely limits the choice of compatible flexible substrates. Water-based inks are compatible with paper [23,24] or with plastic substrates previously coated with a sintering-promoting layer [25]. On the other hand, green organic solvents, such as alcohols, enable adequate wetting of untreated and uncoated plastics such as polyethylene terephthalate (PET) [26] and polyimide (PI) [16], but there are few compatible stabilizers.

In this work, a simple two-step batch process was used to fabricate amphiphilic nanosilver particles. First, the synthesis of AgNPs coated with a hydrophilic stabilizer (PAA) was performed according to our previous work [25]. Second, the primary protective agent was modified by forming amide bonds with a secondary stabilizer (3-morpholinopropylamine) to adjust the hydrophilic–lipophilic balance of AgNPs. While similar approaches of improving the wettability of microparticle inks have been reported [27], particles larger than 200–500 nm are generally considered incompatible with inkjet printers as they cause nozzle clogging [28]. To our knowledge, this is the first example of developing such amphiphilic particles at the nanoscale. The physicochemical properties of the prepared AgNP dispersions were evaluated and fine tuning of the ink chemistry was performed to formulate a conductive ink that fits inkjet printing requirements. After printing on different flexible substrates, the printed patterns were processed using intense pulsed light (IPL) and exhibited excellent electrical properties.

## 2. Materials and Methods

### 2.1. Reagents

All reagents were of analytical grade and solutions were prepared using ultrapure water (Millipore MilliQ system, TKA Germany). Silver nitrate (AgNO_3_) and *N*-methyl-2-pyrrolidone (NMP, 99.9 wt%) were obtained from VWR Chemicals (Leuven, Belgium). Poly(acrylic acid) (PAA, *M*_w_ = 1800), hydrazine hydrate (N_2_H_4_; 50–60 wt%), ethylene glycol (EG; anhydrous, 99.8 wt%) and 3-morpholynopropylamine (MPA) were purchased from Sigma-Aldrich (Burlington, MA, USA). Terpineol (TpOH; mixed isomers, 98 wt%) was purchased from Alfa Aesar (Kandel, Germany). Ethanol (EtOH, 96 wt%), isopropyl alcohol (IPA; anhydrous, 99.5 wt%) and citric acid were obtained from Gram-Mol (Zagreb, Croatia). *N,N′*-diisopropylcarbodiimide (DCI) was bought from Merck (Darmstadt, Germany) and sodium hydroxide from Kemika (Zagreb, Croatia).

### 2.2. Synthesis and Characterization of Powdered Nanosilver Product

Two separate synthesis procedures were performed to prepare silver nanoparticles encapsulated with two different stabilizing agents. First, silver nanoparticles coated with poly(acrylic acid) (PAA-AgNP) were prepared according to [25]. Amphiphilic silver nanoparticles were synthesized as described by Ahn and Lewis for microparticle synthesis [27], with slight modifications to the procedure. Briefly, a mixture of PAA- stabilized silver nanoparticles (4.5 g), NMP (10.0 mL), DCI (4.2 mL) and MPA (4.0 mL) was sonicated for 3 h in a water bath without heating. The dispersion was then refluxed at 60 °C for 20 h. The amphiphilic silver nanoparticles (MPA-PAA-AgNP) were collected after centrifugation at 3000 rpm overnight and dried at 80 °C.

Dynamic thermogravimetric measurements (DTA) of the prepared powdered product were performed using a Q500 (TA Instruments) thermogravimetric analyzer. The measurement conditions were: a heating rate of 10 °C/min, an oxygen flow rate of 60 mL/min and a temperature range of 20–500 °C. Fourier transform infrared spectroscopy (PerkinElmer Spectrum One FTIR spectrometer equipped with ZnSe crystal as an internal reflection element) was used to gather information on the chemical composition of the PAA-capped and MPA-PAA-capped silver powder. Infrared spectra (average of 32 scans recorded for each spectrum) were collected under ambient conditions in the range of 4000–500 cm^–1^ and with a resolution of 4 cm^–1^.

### 2.3. Density Functional Theory (DFT) Calculations

Density functional theory (DFT) calculations were used to elucidate the effect of amidation on the stability of PAA-capped AgNPs in less-polar solvents. For that purpose, the geometries of interacting systems were optimized in Gaussian 16 at the PBE0-D3BJ/def2SVP theory level [29,30], i.e., NMP interacting with PAA (NMP-PAA), the Ag_8_ cluster with PAA (Ag_8_-PAA), and amidated PAA (MPA-PAA) with Ag_8_ (MPA-PAA-Ag_8_). PBE0 amended with D3BJ was chosen for its good predictive capabilities [30,31,32], whereas Def2-SVP was chosen as the basis set for the calculations due to its accuracy and exceptional computational efficiency [33]. An Ag_8_ model cluster was chosen in order to minimize the computational expenses, and at the same time, to provide a reasonable model chemistry. The *D*_2d_ point-group symmetry cluster, akin to that reported by Fournier [34] as one of the most stable Ag_8_ isomers, was obtained via optimization at the aforementioned level of theory. Charged clusters of similar size and size distributions, obtained via reduction of AgNO_3_, have been reported elsewhere in the literature [35]. In addition, it is highly likely that neutral AgNPs containing even-numbered silver atoms are formed during synthesis due to the strong reducing conditions [36].

The effect of NMP as the solvent on the optimized geometries was implemented by using the integral equation formalism variant polarizable continuum model (IEFPCM). The static dielectric constant (*Κ* = 32.3) for NMP was taken from PubChem, while the dynamic dielectric constant (*Κ*_∞_ = 2.1580) was calculated as the square of the refractive index available on PubChem as well [37]. The vibrational frequencies of the optimized structures were also determined, to ensure that the obtained structures represented actual minima on the potential surface and not transition states. Cartesian coordinates of the optimized molecules are given in Appendix A. The counterpoise corrected complexation energies were determined by the approach of Wang and Newton [38]. Gaussview 6 [39] was used to prepare and visualize the input and output files for Gaussian calculations; in addition, Multiwfn [40] was used to perform extended charge-decomposition analysis (ECDA).

### 2.4. Preparation and Characterization of Silver Nanoparticle Dispersions

The as-prepared PAA-AgNPs and MPA-PAA-AgNPs were first redispersed in several pure liquid media of different polarity—water, ethanol, isopropyl alcohol, ethylene glycol and terpineol. Amphiphilic nanosilver particle-based conductive ink with a solid loading of 10 wt% was prepared by redispersing powdered AgNPs in a ternary solvent mixture of water, EtOH and EG in volume ratio of 0.45:0.45:0.1, respectively. A combination of a Sonorex high-power ultrasonic bath (Bandelin electronic, Berlin, Germany) and a Sonoplus 2000.2 ultrasonic homogenizer equipped with an MS-72 sonotrode (Bandelin electronic, Berlin, Germany) was used for sample homogenization.

The absorption maxima and stability of the prepared nanosilver materials dispersed in pure solvent and ink formulation were determined using a Shimadzu UV-1280 UV-Vis spectrometer. The absorption spectra were recorded in a spectral range of 700–350 nm with a specimen dilution of 1:200 by volume; scan rate was set to medium to diminish the spectral noise. The average size and size distribution of PAA-AgNPs and MPA-PAA-AgNPs dispersed in pure solvents and ink formulations were measured via the Dynamic Light Scattering (DLS) method using the Zetasizer Ultra particle analyzer (Malvern Pananalytical, UK). The nanosilver specimen (sample dilution *φ* = 1:200) was placed into a folded capillary cell (DTS1070) and irradiated using an incident He-Ne laser light (*λ* = 632.8 nm); the intensity of the scattered light was converted to the contribution per number of particles within the measured sample volume. Before each measurement, the samples were equilibrated for 120 s at a temperature of 25.0 ± 0.1 °C. The stability of the prepared nanosilver particles with respect to the zeta-potential measurements was determined using the same instrument and samples as for the DLS measurements.

The viscosity of pure solvents and solvent mixtures, as well as simple nanosilver dispersions and conductive ink, was measured using an Ostwald micro-viscometer (516 13/Ic, SI Analytics GmbH, Germany) suitable for small fluid quantities (*V* = 2 mL). Measurements were carried out at a room temperature of 21.5 ± 0.6 °C; the mean value of three consecutive measurements is given. The surface tension of the pure solvents, solvent mixtures and prepared nanosilver dispersions was measured using a Krüss K6 force tensiometer (Krüss Optronic GmbH, Germany), under ambient conditions (*t* = 22.2 ± 0.6 °C); experimental data were temperature-corrected using a correction factor provided by the manufacturer.

Contact angles were measured using the sessile drop method by placing a drop of amphiphilic particle-based conductive ink (10 wt%) onto photographic paper, PET and PI substrates. The photo of the formed drop was taken using the CCD camera of an OCA 20 goniometer (DataPhysics Instruments GmbH, Filderstadt, Germany), and the contact angle was measured within the first 10 s after the drop was formed. The average values of at least three droplets on different flexible surface sites were taken and the standard deviation for all substrates showed very small variation (less than 1°). The measurements were performed at an ambient temperature of 23.6 ± 0.3 °C.

### 2.5. Deposition Process and Post-Printing Treatment

Inkjet printing was performed using an industrial flatbed *Drop-on-Demand* (DoD) printer with an Epson xp600 printhead on glossy photo paper (ORINK, Zhuhai, China), and Novacentrix Novele^TM^ precoated PET (Austin, TX, USA) and PI (*d* = 25 μm, Kapton, DuPont, Wilmington, NC, USA) flexible substrates. A conductive ink based on amphiphilic silver nanoparticles formulated in a glass vial was directly connected to the printer cartridge corresponding to black ink. The inkjet printing process was performed under ambient conditions.

Intense pulsed light (IPL) was used to convert the printed nanosilver pattern into a functional metallic conductor. For that purpose, printed squares were set in a Xenon X-1100 IPL system sample chamber and flashed with a Xenon LH-912 lamp source. A series of experiments were conducted at 2500 V to find the optimal flashing energy for various flexible substrates. Electrical properties were characterized by measuring the surface resistivity (sheet resistance) of the printed tracks before and after IPL processing using a Four-Point Probe (Ossila, UK).

Morphological characterization of the printed silver features on the paper and plastic substrates was evaluated using a scanning electron microscope (TESCAN Vega3 Easy Probe, Brno, Czech Republic) equipped with an energy-dispersive detector (EDS) for chemical composition analysis.

## 3. Results and Discussion

### 3.1. Synthesis of Amphiphilic Silver Nanoparticles; an Experimental and Theoretical Aspect

Amphiphilic silver particles were fabricated via an amidation reaction, in which previously prepared poly(acrylic acid)-protected silver nanograins [25] were modified with 3-morpholinopropylamine. The basic parameters of AgNPs (i.e., particle size and shape) can be controlled with the amount of precursor and capping agent used. It has been shown that a lower stabilizer-to-precursor ratio leads to the formation of bigger particles with a higher polydispersity index, which can be explained by an increased contribution of aggregation in the NPs’ growth [41].

To avoid particle growth, as well as possible nozzle clogging during the printing process, the molar ratio of the added morpholino group to the primary PAA stabilizer was set to 10. The amidation reaction was conducted in NMP, since activation of the carboxylic acid for successful and high-yield formation of an amide bond by the DIC molecule requires the usage of dipolar aprotic solvents [42,43]. NMP is a strongly polar and aprotic solvent, with an exceptional combination of physicochemical properties, making it a suitable dispersion medium for various nanomaterials [44]. By dispersing PAA-stabilized AgNPs in NMP—a starting step in the synthesis of amphiphilic particles—a homogenous dispersion with a characteristic brown-yellowish tone is obtained, indicating good particle stability. In contrast to protic solvents, in which the stability of charged polyelectrolyte-coated nanoparticles is dictated by the formation of repulsive Coulombic forces between ionized carboxyl groups, the high stability of as-prepared PAA-AgNPs in aprotic solvent is determined exclusively by steric interactions, as the carboxyl group cannot ionize. To investigate the intermolecular interactions between PAA and NMP in the NMP liquid phase in more detail, the geometry of the model PAA (C_20_O_14_H_28_, *M*_w_ = 0.49 kDa) and an NMP molecule was optimized. The obtained structure shown in Figure 1 clearly indicates the affinity of NMP to form hydrogen bonds with PAA.

The ability of NMP to form hydrogen bonds is well known, as NMP readily forms hydrogen bonds with water, for example [45], albeit lacking the ability to form hydrogen bonds with itself. It is therefore not surprising that calculations have revealed that NMP readily forms a hydrogen bond with PAA via its ketone group and the protonated PAA molecule. The favorable intermolecular interactions are further determined by the favorable complexation energy of –57.1 kJ mol^–1^, which is in reasonable agreement with the strength of hydrogen bonding reported in the literature [46]. Thus, NMP is a suitable solvent or dispersant for PAA-capped AgNPs.

To assess the interactions and dipole moments of the PAA/Ag_8_ and MPA-PAA/Ag_8_ model systems in NMP, DFT calculations were performed. It can be assumed that PAA is likely chemisorbed to the AgNP surface by a donor-type bond formed during synthesis and that this bond remains unchanged during subsequent amidation, especially since the formation of water as a leaving group is necessary during amidation. The calculated interaction energies (Appendix A) indicate that MPA-PAA has even higher complexation energy with the Ag_8_ cluster in NMP than PAA alone. ECDA revealed that PAA alone donates 0.64 electrons to the Ag_8_ cluster, while MPA-PAA donates 0.89 electrons due to an additional charge transfer occurring from the morpholino group, as shown in the molecular orbital contour plots in Figure 2a,b. Moreover, electrostatic potential (ESP) mapping shown in Figure 2c–e clearly indicates that the Ag_8_ cluster is more negatively charged in the case of MPA-PAA/Ag_8_, thus leading to a favorable increase in complexation energy. In both cases, the point group symmetry of the Ag_8_ nanocluster in the optimized complexes with PAA and MPA-PAA decreases from *D*_2d_ to *C*_2v_. The determined dipole moment of 9.31 D for the PAA/Ag_8_ system is significantly smaller than the value of 22.23 D determined for the MPA-PAA/Ag_8_ system. The larger dipole moment magnitude leads to improved dispersion stability of the MPA-PAA-capped AgNP in polar solvents, such as NMP, over PAA-capped alone. This indicates that modification with a less-polar MPA moiety, which should improve the wetting of plastic substrates, does not reduce the dispersion stability of the AgNPs in polar solvents (as confirmed in Section 3.3.1.).

To gain insight into the properties of the prepared nanosilver material, the absorption spectra of AgNPs in the reaction mixture were acquired before and after amidation.

Colloidal dispersions of silver nanoparticles scatter optical light efficiently due to the surface plasmon resonance (SPR) effect, i.e., the collective resonance of the conduction electrons. The spectrum bandwidth, magnitude and SPR wavelength peak associated with the nanoparticles strongly depend on the particle size, shape and environment. A sharp SPR maximum around 420 nm indicates spherical silver particles with a small hydrodynamic radius; a narrow full width at half maximum indicated monodisperse particle distribution [9]. The absorption bands for PAA-AgNPs and MPA-PAA-AgNP are of similar size and shape (Figure 3); the intensity of the SPR band (*λ*_max_ = 422 nm) is slightly more pronounced for polyacid-coated particles than for the amphiphilic particles. The absorption maximum for amphiphilic particles is marginally red-shifted (*λ*_max_ = 424 nm), indicating that slightly larger particles are obtained. This is not surprising since the amphiphilic particles were prepared via amidation of the primary particle stabilizer, resulting in the formation of a larger organic shell encapsulating the particles. This also causes a reduction in peak intensity.

### 3.2. Characterization of the Powdered Silver Material

Due to their applications in printed electronics, i.e., the need to remove the protective organic layer and achieve the best electrical properties, thermal stability is an important property of AgNPs. In order to distinguish the amount of polymeric stabilizer (primary or secondary), thermogravimetric analysis (TGA) was performed (Figure 4).

TGA curves for both PAA-AgNPs and MPA-AgNPs exhibited stepwise mass losses consistent with removal of the protective polymer coating. As shown in Figure 4a, the PAA-stabilized silver nanoparticles began to thermally decompose at 120 °C (decarboxylation of the polyacid) and showed a maximum weight loss around 230 °C (complete decomposition of the polymer backbone) [47]. The total weight loss was 2.3 wt%, which corresponds to the loss of the polyacid stabilizer chemisorbed on the nanograin surface. In the case of amphiphilic silver particles (Figure 4b), three distinct temperature-dependent steps are evident—the first one, below the temperature of 100 °C, corresponding to the degradation of the external morpholino moiety; the second mass loss, observed between approximately 110 °C and 220 °C, attributed to residual NMP; and the third one, above 220 °C, attributed to the decomposition of the primary-bound polyacid stabilizer. Complete removal of the protective organic shell is evident at temperatures above 300 °C, leaving a metallic silver residue with a metal content of 93.25 wt%.

Fourier transform infrared (FTIR) analysis was used to identify the stabilizer attached to the particle surface [48]. Alterations in the characteristic pattern of absorption bands clearly indicate a change in material composition. The FTIR spectra of the as-prepared PAA-AgNPs and MPA-PAA-AgNPs are shown in Figure 5.

The main bands in the PAA-AgNP spectrum appear in the vibrational modes of hydroxyl (–OH, ~3500 cm^–1^), carbonyl (–C=O, ~1700 cm^–1^), and carboxylic acid (–C–O, ~1100 cm^–1^) moieties [49]. In addition, the absorption peak at 2915 cm^–1^ is attributed to the methyl/methylene (–CH_x_) group of the polymer backbone [50]. Except for changes in band intensity and relatively minor broadening of some bands, the fundamental frequencies in the FTIR spectra of PAA-AgNPs are also visible in the spectra of amphiphilic particles. An intense peak at 3445 cm^–1^, the same wavenumber as for particles stabilized solely with PAA, is attributed to *ν*(–OH) stretching, suggesting the presence of a functional carboxylic group of the unreacted polyacid. The enhancement of the band intensity can be ascribed to the superposition phenomena, i.e., the appearance of a new absorption band, visible as a shoulder at 3262 cm^–1^. This corresponds to the –NH stretching vibrations of the secondary aliphatic amine, indicating the successful formation of an amide bond [51]. The introduction of a nitrogen-containing heterocyclic group is also confirmed by the presence of an intense absorption maximum at 1088 cm^–1^, attributed to the *ν*(C-N-C) vibrations of the morpholino moiety. The bands observed at 1629 cm^–1^ and 1698 cm^–1^ are attributed to the *ν*(C=O) and *ν*(CO-NH) vibrations of the newly formed amide bond. The significant peaks observed in the fingerprint region of the AgNP spectra are ascribed to the N-H twisting vibration (band at 791 cm^–1^ for amphiphilic particles) and a –C-C– polyacid skeleton vibration (peak at 608 cm^–1^ visible in both FTIR spectra).

### 3.3. Dispersion Stability and Conductive Ink Formulation

#### 3.3.1. Influence of the Solvents on the Silver Nanoparticle Dispersion Stability

To formulate the inkjet-printable conductive ink, the powdered nanosilver product is usually redispersed in a suitable solvent or solvent mixture [52]. The solvent needs to act as a carrier medium for the ink’s functional material and have the ability to dissolve the polymer without affecting the conductive nanoparticles or substrate. Important requirements for the solvent also include suitability for large-scale production and environmental friendliness [53]. Due to their high surface energy, it is generally difficult to disperse silver nanoparticles in a specific solvent. Therefore, the selection of a suitable carrier medium for long-term dispersion stability is a challenge. In order to examine the influence of the solvent in more detail, the as-prepared PAA-AgNP and MPA-PAA-AgNP powdered products were redispersed in several pure liquid media of different polarity—water (*ε* = 80.4), ethylene glycol (*ε* = 37.0), ethanol (*ε* = 24.5), isopropyl alcohol (*ε* = 18.2) and terpineol (*ε* = 2.8).

To evaluate the tendency of AgNPs to aggregate and eventually precipitate, particle size distribution in the simple dispersion was selected as one control parameter, along with electrokinetic or zeta-potential measurements. It is generally accepted that absolute zeta-potential values greater than ±25 mV are indicative of nanosilver dispersion stability [54]. Therefore, the size and size distribution of PAA-AgNPs (Appendix A–c) and PMA-PAA-AgNPs (Appendix A–e) were investigated via DLS, which further corroborated the zeta-potential measurements. For clarity, the data obtained are summarized in Table 1.

The obtained particle size (expressed as average hydrodynamic diameter or *Z*-value) for PAA-coated AgNPs redispersed in water, EG and EtOH is suitable for application in inkjet printing technology. Moreover, the zeta potentials of the polyacid-encapsulated AgNPs exhibited considerably more negative values, indicating excellent electrostatic stability in those three solvents right after dispersion (Table 1). In the case of IPA and TpOH, the PAA-AgNPs began to precipitate right after dispersion, so their particle size and zeta potential was not determined. In the case of MPA-PAA-AgNPs, we can see that amidation did not significantly affect the size of the particles in those solvents. This finding is important because the synthesis of monodisperse particles with sizes below 50 nm is necessary for the preparation of inkjet-printable conductive ink for the additive nanomanufacturing of flexible electronics [55]. Thus, in terms of nanoparticle size, simple dispersions of amphiphilic AgNPs in pure water, EG and EtOH are suitable for the inkjet printing process. A significant increase in the average *Z*-value for dispersions in IPA and TpOH indicates extensive particle aggregation. The electrokinetic potential of the amphiphilic silver particles redispersed in all solvents except TpOH exhibited highly negative values. This could be due to the presence of a hydrophobic morpholino group, which forms a surface barrier and, thus, reduces or completely prevents interparticle interactions, i.e., creates a sterically stabilized system [41]. However, the negative electrokinetic potential may also be due to the presence of the unreacted polyacid component, which is evident in the FTIR analysis.

To evaluate long-term stability, UV-Vis measurements of the prepared dispersions were conducted over a period of one month (Figure 6).

In the case of AgNPs encapsulated with PAA, the overall greatest stability was achieved in water, due to the strong electrostatic repulsion from the ionization of PAA. Within the first two weeks after preparation, there is no significant change in the shape and position of the absorption bands of the most polar solvents, H_2_O and EG (Figure 6a,b). As time progresses, the SPR bands broaden until finally, a secondary maximum (shoulder) appears, which leads to a significant decrease in the primary SPR intensity. This indicates a decrease in the particle density, i.e., the growth and precipitation of silver particles [52]. Ethanol is a poor solvent for poly(acrylic acid) and induces partial precipitation of PAA-AgNPs particles [56]. In UV-Vis spectra, this is manifested through decreasing SPR band intensity and through the appearance of secondary maxima at higher wavelengths (Figure 6c), although this dispersion remained a yellow-brownish color after one month (Appendix A). Shortly after preparation, the dispersion of nanoparticles in IPA began to precipitate to a greater extent than in EtOH (Figure 6d). It can also be seen in all the graphical representations (Figure 6a–d) that the SPR absorption band of the AgNPs gradually shifts to shorter wavelengths. These observations can be explained by the gradual formation of a thin oxide layer on the surface of the nanograins [57]. The dispersions prepared in TpOH became unstable as the hydrophilic silver particles precipitate after only one day.

Better stability of amphiphilic particles in less polar solvents was expected as the primary stabilizer (PAA) was modified by a primary amine, providing strong steric stabilization. Additionally, indeed, the stability of the amphiphilic particles in the same pure solvents was found to be better than that of the polyacid-protected AgNPs (Figure 6 and Appendix A). In general, the SPR bands of the amphiphilic particles dispersed in water, EG, EtOH and IPA retained their shape and position, without the formation of additional absorption maxima. A significant improvement in stability over PAA-AgNPs is observed in ethanol and isopropyl alcohol dispersions, where the formation of the shoulder is negligible (Figure 7c,d). These results are in accordance with those obtained by DFT. Similar to PAA-AgNPs, the UV-Vis spectra of amphiphilic particles dispersed in pure TpOH indicate colloidal instability (Figure 7e). Compared to other simple dispersions, the shape of the SPR band broadens immediately after sample preparation, and the intensity decreases. The nanoparticles aggregate and are completely precipitated two weeks after preparation.

#### 3.3.2. Influence of Silver Nanoparticles on Dispersion Viscosity and Surface Tension

The size and morphology of the nanograins, as well as the mass fraction of the particles, strongly affects the rheology of the ink due to the complex solvent–particle or particle–particle interactions, and this can, in turn, affect inkjet printing. Despite numerous studies, to date, there is no single theoretical model to predict the rheological behavior of nanoparticle-based inks [58]. Therefore, the viscosity and surface tension of dispersions of nanoparticles in different carrier media—the key printing parameters that enable optimal printing performance—are determined experimentally.

The viscosity of a simple dispersion composed of spherical nanoparticles with a volume fraction of <10% is linearly proportional to the solid content. Such dispersion is classified as a Newtonian fluid and can be described by the Einstein equation [59]. The viscosity of 1 wt% AgNP dispersions was imperceptibly higher than the viscosity of pure solvents for particles encapsulated with both polymeric layers (Figure 8a,b). It can be concluded that the viscosity of the simple dispersion of nanosilver particles both before and after amidation is determined more by the viscosity of the pure solvent than by the capping agent used.

Surface tension plays an important role in any droplet formation processes, including inkjet printing. The surface tension of pure solvents examined as a carrier medium for amphiphilic nanosilver particles ranged from 72.4 mN m^–1^ (water) to 22.15 mN m^–1^ (IPA), respectively (Figure 8c,d). Similar to viscosity, the measured surface tension values changed insignificantly after the addition of 1 wt% nanosilver particles before and after amidation. In general, the mechanism of surface tension in dispersions composed of nanomaterials is a complex phenomenon that depends on several parameters such as particle size and geometry, particle surface coverage, base fluid, and the mass fraction of nanoparticles at the fluid/gas interface [60]. Experimental studies published in the literature have shown contradictory trends in surface tension as a function of nanoparticle content. A decrease in the surface tension of nanosilver fluid with low ionic strength (particles dispersed in distilled water) was reported [61,62], and is in agreement with our results. This may be due to the large spacing between nanoparticles in the dilute dispersions (1 wt% of nanosilver material), which favors electrostatic forces between PAA-AgNPs, or static repulsions between MPA-PAA-AgNPs. On the other hand, the surface tension of alcoholic simple dispersions was found to increase for both PAA-AgNPs and MPA-PAA-AgNPs compared to pure solvents. An increase in the surface tension of colloidal silver dispersions with increasing nanosilver solid loading has been reported in the literature for aqueous diethylene glycol solutions [59], and aqueous IPA solutions [63]. In addition, Fernandes et al. [7] reported surface tension changes in dispersions with the same nanosilver wt% and different volume ratios of the solvents used as the carrier medium, indicating that the dispersions with lower AgNPs wt% had higher surface tension. Therefore, for inks intended for inkjet printing, it is mandatory to determine the surface tension experimentally, regardless of their composition.

#### 3.3.3. Formulation and Characterization of the Conductive Ink Based on Amphiphilic Particles

For successful inkjet printing, there are strict restrictions regarding the ink viscosity and surface tension; the optimal values are in the range of 2–6 mPa sand 30–34 mN m^–1^, respectively [59]. We see from Figure 8 that dispersions of silver nanoparticles in pure solvents cannot serve as a functional ink since their viscosity is either too low (water, EtOH and IPA), or too high (EG and TpOH). To overcome this obstacle, co-solvents are usually added to improve the ink’s compatibility with the printing system, and/or to modify its drying properties. Additionally, an undesirable coffee ring effect can be reduced or avoided by using a binary solvent mixture, one with a high boiling point and low surface tension and the other with a high surface tension and low boiling point. Since viscosity and surface tension are dictated by the solvent (Figure 8), we characterized binary, ternary and quaternary water–alcoholic mixtures to find a solvent base with optimal properties (Appendix A). Finally, we settled on a ternary H_2_O/EtOH/EG mixture with 10 wt% of MPA-PAA-AgNPs. The as-prepared conductive ink exhibited satisfactory viscosity of 2.51 mPa s and surface tension of 31.26 mN m^–1^, respectively.

The size, satisfying dispersion and high colloidal stability of silver nanoparticles are crucial for the formulation of conductive ink to avoid nozzle clogging due to the use of large particles or the agglomeration of smaller particles in an insufficiently stable colloid. Thus, the size distribution of amphiphilic particles in the ink formulation was determined (Figure 9a); the average hydrodynamic radius was 37.96 nm, with a corresponding electrokinetic potential of –33.35 mV, indicating satisfactory size and stability upon preparation. The long-term stability of the conductive ink was investigated by monitoring the SPR absorption band of the amphiphilic particles over a period of one month (Figure 9b).

It is obvious that the absorption band maintains its size and shape during the period of one month. The UV-Vis spectrum has a symmetrical shape with a narrow full width at half maximum, indicating good particle stability. The slight decrease in the absorption SPR intensity could be due to the formation of a thin silver film—a silver mirror—on the walls of the glass vial in which the ink was stored.

### 3.4. Inkjet Printing and IPL Post-Print Processing

Adhesion between the conductive ink and the substrate must also be exceptional to obtain satisfactory electrical properties for printed electronics [4]. The adhesion depends on both the morphological and chemical nature of the interface, and can be improved by modifying either the ink chemistry or the substrate surface. PAA-AgNPs showed inhomogeneous wetting on untreated plastic surfaces and poor adhesion. The amphiphilic nanoparticles were encapsulated in an organic corona layer, where the precise ratio of hydrophilic and hydrophobic groups is believed to provide good adhesion to substrates with different surface energy. Polymeric plastics such as PET, polyethylene naphthalate (PEN) and PI are commonly used to manufacture flexible printed electronics. In addition to plastics, paper is also widely adopted as a flexible substrate for the fabrication of sustainable electronics due to its low cost, natural origin, abundance, biocompatibility and biodegradability [64]. However, the porous cellulose paper network, together with the relatively high surface roughness, tends to absorb the liquid ink and create pinholes of the deposited conductive material. Therefore, to prevent failure of the functional device, plastic coatings are used in many examples of paper electronics [11,63].

We decided to test the printability of the prepared amphiphilic AgNP conductive ink on three flexible substrates with different composition, thermal stability and surface energy—glossy paper and two polymer foils. The paper and PET substrate used in this study were coated with an adhesion-promoting layer that allows the ink to penetrate and be more stable, whereas the PI surface remained untreated. Before printing, the surface wettability of the prepared ink with respect to each substrate was investigated by measuring the contact angles. In general, low contact angles indicate good wettability, while high contact angles indicate poor wettability [65]. The contact angles of the silver ink formed with the flexible substrates were: 19.95 ± 0.75° on photopaper, 19.55 ± 0.77° on coated PET and 24.9 ± 0.55° on PI, respectively (Appendix A). The obtained values indicate that the amphiphilic silver dispersion wets all three substrates well and that favorable adhesion is obtained. The measured contact angles for amphiphilic silver particles are lower than those published in the literature [66], especially for PET foil, indicating better wettability [67,68]. Multiple layers (two, three and four) were printed to reduce electrical resistance, a common strategy in printed electronics [69]. Four layers proved to be optimal to achieve excellent electrical properties on all flexible substrates.

To convert the printed layer into a functional conductive pattern, a post-processing step is essential. In theory, this may simply involve evaporation of the solvent and other ink additives, but in practice, a more complex sintering process is usually required to achieve sufficient conductivity [70]. Conventional thermal heating is usually detrimental to most plastic substrates used in flexible electronics. It is also a time- and energy-consuming process that is not appropriate for low-cost and high-speed device fabrication. Therefore, alternative sintering processes must be used to enable fast and selective sintering of materials. By using broad-spectrum intense pulsed light (IPL), encompassing UV/Vis/IR, the energy supplied in a pulse can be matched to the energy required for the stabilizer to degrade, furthermore selected locations on a sample can be targeted [71]. This causes the removal of organic stabilizers and the generation of conductive necks in the printed sample, resulting in the formation of a homogeneous conductive line. Another advantage of IPL is the reduction in sintering time from hours/minutes (thermal sintering) to milliseconds or even microseconds [72].

To reduce the immense initial sheet resistance value, we subjected the printed geometries on all flexible substrates to an IPL annealing process. Due to their small size and low heat-treatment temperature, AgNPs are promising candidates for a successful IPL sintering process [73]. To determine a suitable energy density for IPL sintering, different energies (range of 300–1000 J) for different substrates were tested by irradiating inkjet-printed amphiphilic silver ink squares in single-pulse mode. To gain a detailed insight into the effect of photo-annealing on the printed silver patterns, morphological characterization was performed using SEM-EDX analysis.

Due to the precoating layer, which facilitated chemical sintering at room temperature, the silver squares printed on a glossy paper substrate were slightly conductive immediately after printing, and the measured sheet resistance was 41.3 kΩ sq^–1^. This result may be due to the surface roughness, as paper is a highly porous and fibrous substrate in which large fibers can cause nanoparticle separation and, thus, have an insulating effect. The effect of photo-annealing on coated paper was remarkable; sheet resistance values ranging from 3.85 Ω sq^–1^ to 6.58 Ω sq^–1^ were obtained at pulsed light energy in the range of 300–700 J (Figure 10). The lowest sheet resistance value was obtained when the thin film was irradiated with 600 J; thereafter, a slight increase in sheet resistance values is observed with a further increase in IPL energy. This can be attributed to the formation of a silver oxide layer on the surface of the printed features.

The SEM micrographs presented in Figure 11 reveal that a partially homogeneous conductive silver layer was obtained on the paper substrate. The thin film was also partially interspersed with larger cracks and holes, which reflected high initial resistance (Figure 11b). After exposing of the printed geometries to IPL flashing, a continuous and homogeneous silver film with negligible cracks was observed (Figure 11c). In addition to silver, the EDX spectra revealed the presence of oxygen and alumina (Figure 11d), while EDS mapping reflected the uniform distribution of Ag, O and Al elements in the printed patterns (Figure 11e). The presence of oxygen and aluminum atoms is attributed to the top coating substrate layer [74].

The SEM image of the amphiphilic silver particles inkjet-printed on a coated PET substrate (Figure 12a) before sintering treatment shows an inhomogeneous silver film, with visible thinner (dark grey) and thicker (bright grey) areas (Figure 12b). After the inkjet-printed silver feature was IPL-flashed with an energy of 200 J (higher energies resulted in detachment of the silver film and were detrimental), the measured sheet resistance decreased from an initial value of 29.18 Ω sq^–1^ to a final value of 0.57 Ω sq^–1^. This value is the lowest obtained with the IPL sintering of the inkjet-printed amphiphilic particles studied in this work. These results are consistent with the microscopic images, showing the formation of a more homogeneous metallic film (Figure 12c). Moreover, the EDX spectra (Figure 12d) confirmed the presence of silver, oxygen and silicon. The distribution of silver atoms, determined via EDX elemental mapping (Figure 12e), is consistent with the thinner and thicker regions of the printed silver pattern. The presence of silicon could be due to the PET coating layer, while oxygen and carbon may originate from the flexible foil itself.

Due to the lack of an adhesion-promoting layer, printing on an unmodified PI substrate is the most challenging [8]. Immediately after printing and drying, the thin silver film was not conductive (Figure 13a) and SEM analysis revealed surface topography consisting of smaller and larger voids (Figure 13b). After IPL sintering (energy of 700 J), interconnections between the amphiphilic silver particles were observed, resulting in the formation of a partially homogeneous film which showed areas of thinner and thicker silver coating (Figure 13c–e). Nevertheless, the printed squares became electrically conductive, exhibiting a low sheet resistance value of 19.71 Ω sq^–1^. To demonstrate the applicability of printed electronics obtained in this manner, an institution logo (University of Zagreb, Faculty of Chemical Engineering and Technology) was printed (4 layers) on PI substrate, which was treated with IPL, and a light-emitting diode (LED) was pasted in the logo center (Figure 14). Applications of flexible electronics require that the printed conductive features operate even when bent. It can be seen that the LED lit up in both cases (planar and bent) as soon as the conductive circuit was connected to the single 9 V battery. The successful fabrication of the functional circuit provides a good basis for the future fabrication of flexible electronics via inkjet printing.

The electrical properties of inkjet-printed nanosilver patterns reported in the literature vary with the size of the particles and ink formulation, as well as with the number of printed layers, sintering method and time used. Appendix A summarizes the main parameters of various silver flexible electronics fabricated via inkjet-printing. It is evident that the electrical features of the thin film of amphiphilic silver particles are comparable to or better than those reported in the literature.

## 4. Conclusions

In summary, an easy-to-use method for direct amide bond formation in anhydrous solvent catalyzed by DCI was applied to prepare amphiphilic silver nanoparticles suitable for the fabrication of printed electronics using inkjet printing technology. To prepare a nanosilver colloid with optimal physicochemical and rheological properties, dispersions of the as-prepared AgNPs in various solvents were investigated in terms of dispersibility, stability, particle size and size distribution, viscosity, and surface tension. In addition, thermal behavior was determined and qualitative analysis performed via TGA and FTIR analysis. A conductive ink containing 10 wt% amphiphilic particles was prepared for printing purposes, with an average particle size of 37.96 nm, zeta-potential value of –33.35 mV, and suitable rheological properties. The conductive ink was inkjet-printed on three flexible substrates (paper, PET and PI) and given highly conductive features using broadband high-intensity light. The effect of photo-annealing was visible on all three flexible substrates used, and resulted in a measured sheet resistance of 3.85 Ω sq^–1^ (glossy paper), 0.57 Ω sq^–1^ (coated PET) and 19.71 sq^–1^ (bare PI), respectively. The ink development strategy presented here can be further extended to other metallic nanoparticle-based inks for inkjet printing of flexible electronics.

## Figures and Tables

**Figure 1 nanomaterials-12-04252-f001:**
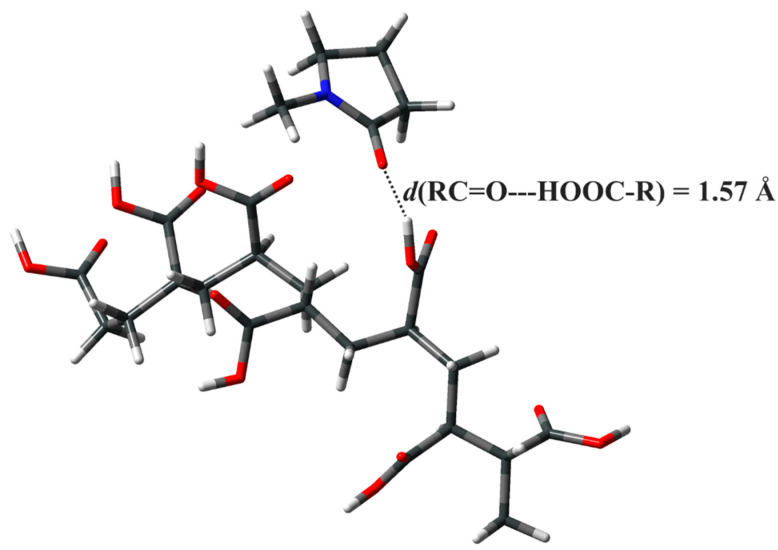
O-HO distance in the optimized complex of PAA and NMP.

**Figure 2 nanomaterials-12-04252-f002:**
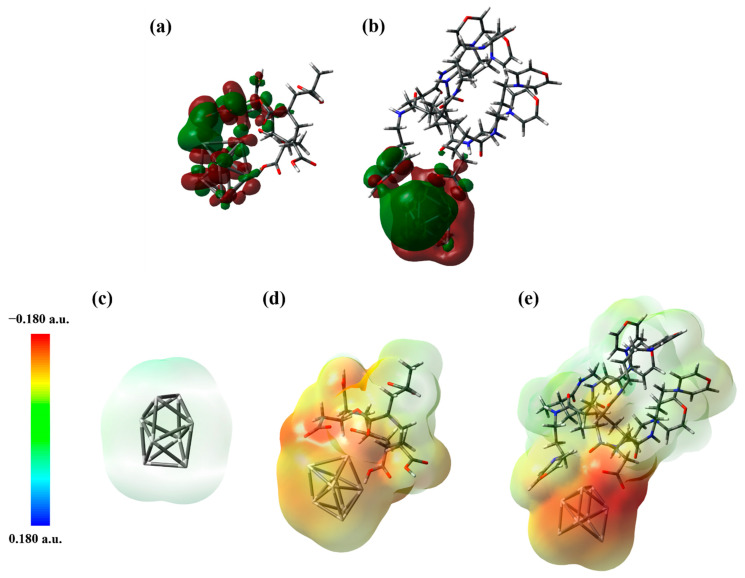
Contour plots (isovalue: 0.01 e Å^−3^) of molecular orbitals with the largest bonding contribution for (**a**) PAA/Ag_8_ (MO = 207), (**b**) MPA-PAA/Ag_8_ (MO = 398) and electrostatic potential (ESP) mapping of (**c**) Ag_8_ cluster, (**d**) PAA/Ag_8_ and (**e**) MPA-PAA/Ag_8_.

**Figure 3 nanomaterials-12-04252-f003:**
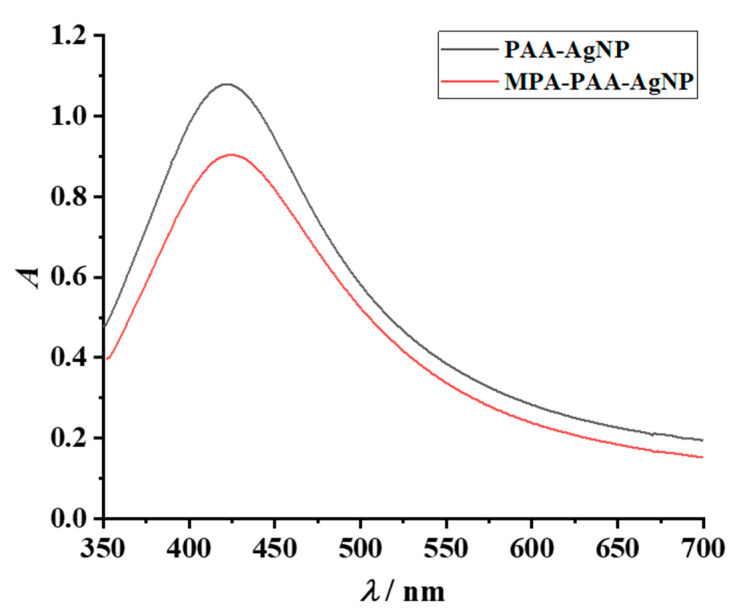
SPR bands for PAA-AgNPs and MPA-PAA-AgNPs, obtained immediately after synthesis with a specimen dilution of *ϕ* = 1:200.

**Figure 4 nanomaterials-12-04252-f004:**
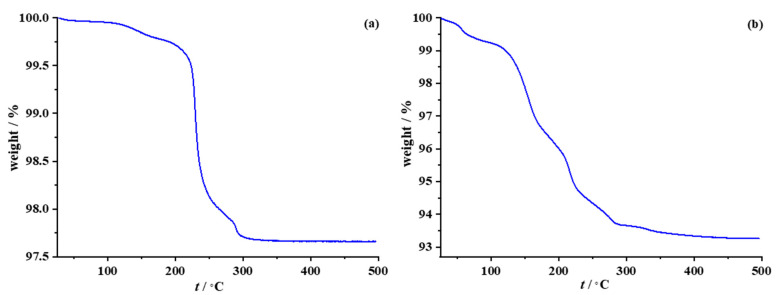
Thermal behavior of (**a**) PAA-AgNP and (**b**) MPA-PAA-AgNP.

**Figure 5 nanomaterials-12-04252-f005:**
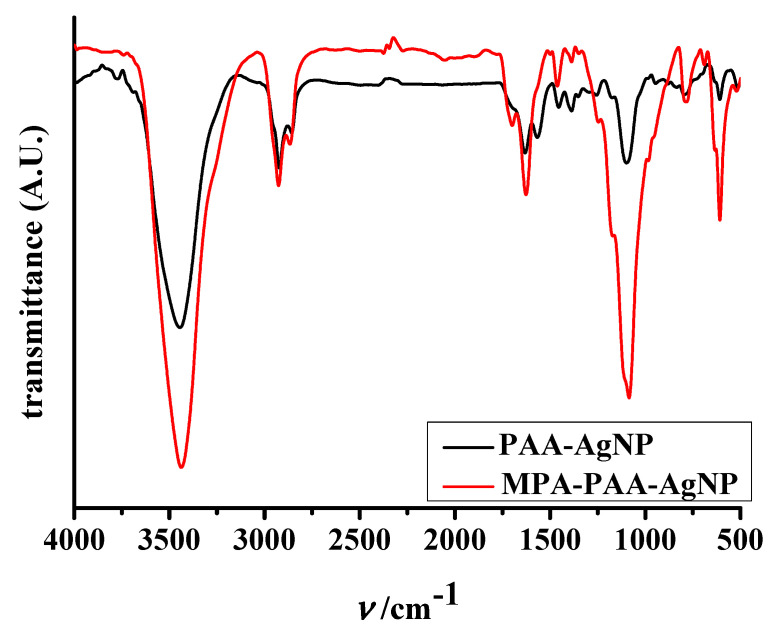
FTIR spectra of PAA-AgNPs and MPA-PAA-AgNPs, respectively.

**Figure 6 nanomaterials-12-04252-f006:**
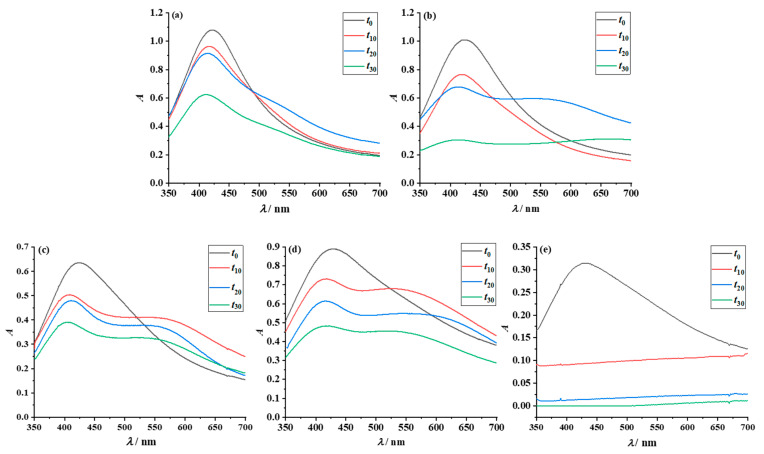
UV-Vis spectra recorded for PAA-AgNPs dispersed in: (**a**) water; (**b**) EG; (**c**) EtOH; (**d**) IPA; and (**e**) TpOH, over a 30-day period.

**Figure 7 nanomaterials-12-04252-f007:**
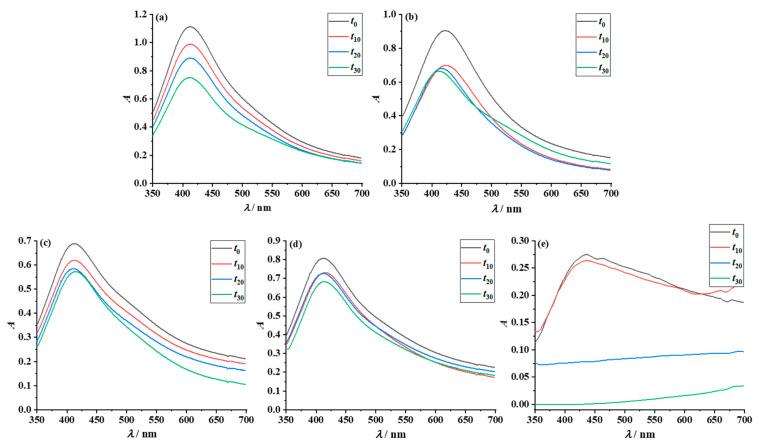
UV-Vis spectra recorded for MPA-PAA-AgNPs dispersed in: (**a**) water; (**b**) EG; (**c**) EtOH; (**d**) IPA; and (**e**) TpOH, over a 30 day period.

**Figure 8 nanomaterials-12-04252-f008:**
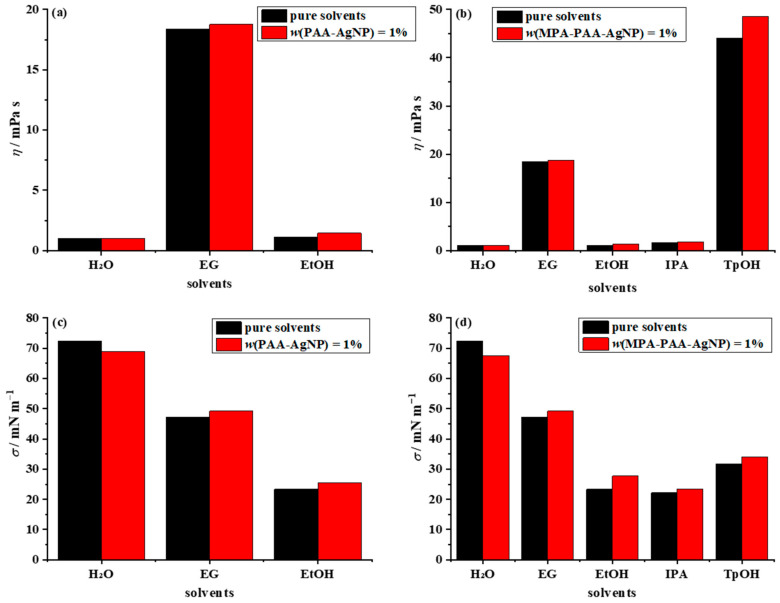
Dispersions of PAA-AgNP and MPA-PAA-AgNP in different carrier solvents and their physical parameters: (**a**,**b**) viscosity and (**c**,**d**) surface tension.

**Figure 9 nanomaterials-12-04252-f009:**
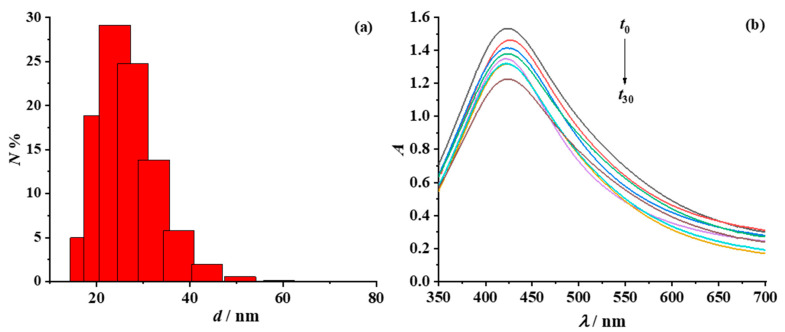
(**a**) Particle size distribution for conductive ink with 10 wt% amphiphilic silver particles and (**b**) the corresponding UV-Vis absorption spectra, recorded during a testing period of one month.

**Figure 10 nanomaterials-12-04252-f010:**
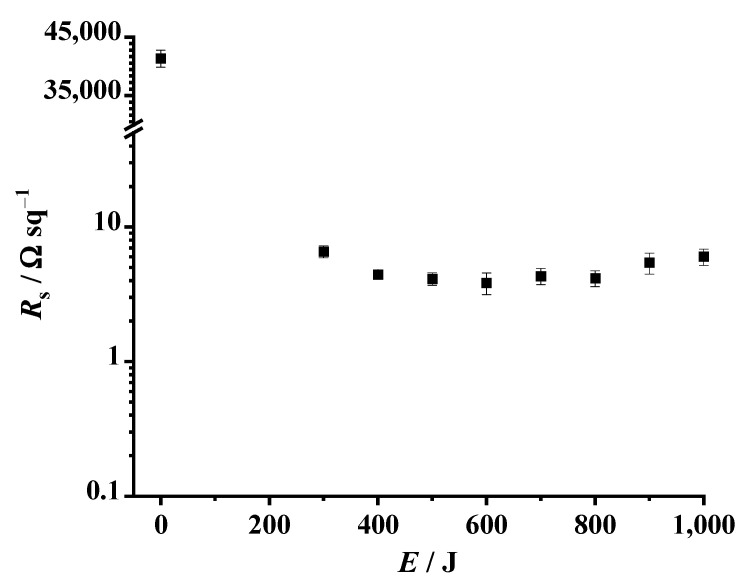
The sheet resistance of the printed silver film on a glossy paper substrate; the error bars represent standard deviation (*n* = 6).

**Figure 11 nanomaterials-12-04252-f011:**
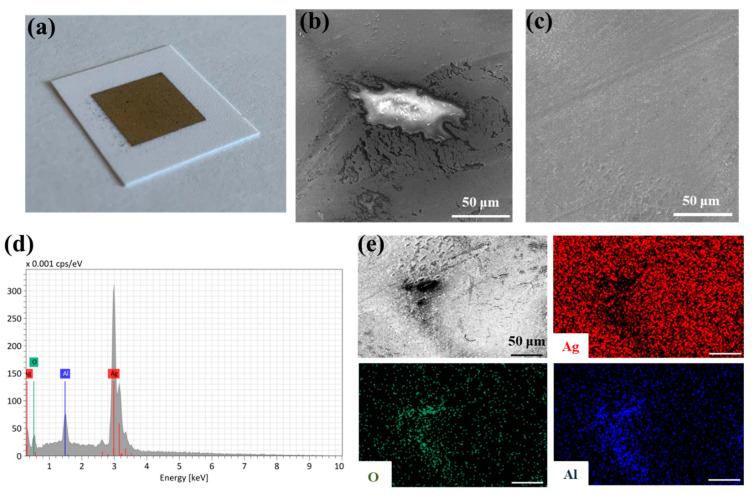
(**a**) Image of the printed silver feature on a glossy paper substrate before annealing, and SEM micrographs of the thin-film surface (**b**) before and (**c**) after IPL annealing; (**d**) EDX spectra and (**e**) elemental mapping of the as-prepared conductive features. Magnification 1000×; scale bar in all micrographs is 50 μm.

**Figure 12 nanomaterials-12-04252-f012:**
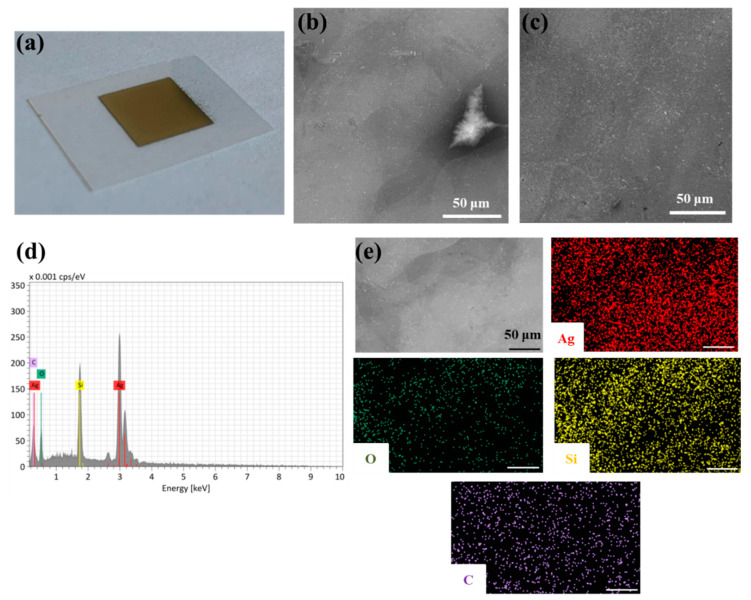
(**a**) Image of the printed silver square on a coated PET substrate before annealing, and SEM micrographs of the thin-film surface (**b**) before and (**c**) after IPL annealing; (**d**) EDX spectra and (**e**) elemental mapping of the prepared conductive features. Shown at 1000× magnification; scale bar 50 μm.

**Figure 13 nanomaterials-12-04252-f013:**
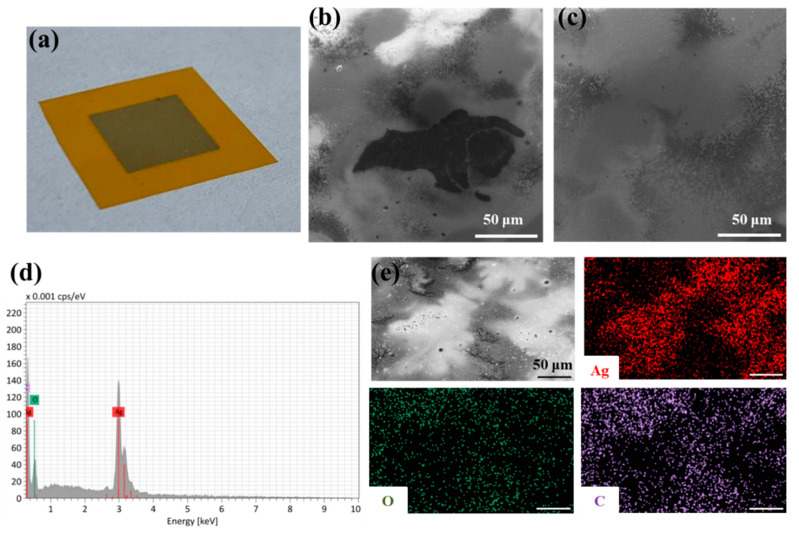
(**a**) Image of the printed silver feature on a bare PI substrate before annealing, and SEM micrographs of the thin-film surface (**b**) before and (**c**) after IPL annealing; (**d**) EDX spectra and (**e**) elemental mapping of the as-produced conductive squares. Imaged at 1000× magnification; all insert scale bars are 50 μm.

**Figure 14 nanomaterials-12-04252-f014:**
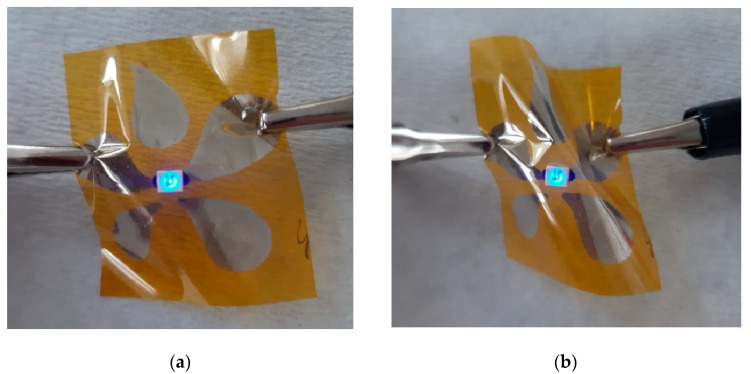
Institutional logo as an example of printed flexible electronics in (**a**) planar and (**b**) bended mode.

**Table 1 nanomaterials-12-04252-t001:** Average particle diameter and corresponding zeta-potential values for PAA-AgNPs and MPA-PAA-AgNPs in simple dispersions.

Solvent	PAA-AgNP		MPA-PAA-AgNP	
	*d*/nm	*ζ*/mV	*d*/nm	*ζ*/mV
H_2_O	39.25	–43.3	34.35	–37.5
EG	36.75	–41.9	37.28	–41.7
EtOH	54.12	–39.0	45.30	–34.3
IPA	–	–	85.65	–32.1
TpOH	–	–	110.30	–21.6

## Data Availability

The data presented in this study are available in the Article and Supplementary Material.

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
