# Peer review of "Amphiphilic Silver Nanoparticles for Inkjet-Printable Conductive Inks"

_nanomaterials, 2022, doi:10.3390/nano12234252_

Round 1

Reviewer 1 Report

1. More comparison work including the particle size, conductivity and sintering parameters or sintering methods should be conducted.

2. The DFT study is very interesting, in which how was the Ag superlattice established? The stable lattice parameters after modeling should be given.

3. Were the organics removed after sintering? What is the content?

4. Could the authors provide some images of the Ag particles before sintering?

Author Response

We thank the reviewer for the comments. Please see the attachment.

Reviewer 2 Report

In the work by Ivanišević et al, the authors characterize several Ag ink formulations for printed electronics. The work is well written, and the characterizations are thoroughly described but it lacks novelty.

A few comments:

-       I suggest to add in the text a comparison between the inks studied and the commercial available ones. Moreover, how is the roll-to-roll technique compared with the inkjet printing?

-       In figure 1 the authors show the formation of a hydrogen bond between PAA and NMP: how the bond is affected when moving from the single PAA residue in Figure 1 to the complex reported in figure 2a?

-       Why the Ag NPs aggregate so fast when in terpineol solution?

-       Why did the authors test a 1 wt% of Ag NPs to select the best solvent (Figure 6, 7 and 8) and then they selected 10 wt% as standard for the final formulation?

Author Response

(The authors gave the same response as above.)

Reviewer 3 Report

In this manuscript, the authors used a secondary stabilizer (3-morpholinopropylamine) to adjust the hydrophilic-lipophilic balance of PAA-coated AgNPs, making it a well-dispersed ink for inkjet printing technology. The physicochemical and rheological properties, dispersions of the as-prepared AgNPs in various solvents were systematically investigated. I would suggest the authors consider my following comments and suggestions.

1.      SEM images of the as-prepared MPA-PAA-AgNPs should be given to provide a visual illustration of the particles’ size distribution.

2.      The authors investigated the inks’ surface tension in figure 8. When dispersing MPA-PAA-AgNPs in water, the inks’ solvent decreases, but the tendency for other solvents is the opposite. The behavior is different when compared with your cited ref [26] (Mater. Chem. Phys.,2014, 148, 686-691, Figure 4a). Could you explain the results? Does the surface tension variation in solvents reflect the hydrophilic-lipophilic balance of MPA-PAA-AgNPs.

3.      Contact angle of MPA-PAA-AgNPs-based inks should be measured to illustrate the ink’s surface tension and wettability during the printing process.

4.      The MPA-PAA-AgNPs are dispersed in a ternary H2O/EtOH/EG mixture. When EtOH is added to the ink, an outward Marangoni flow would be induced, which counterbalances the inward flow boosted by the addition of EG, and leads to the coffee ring effect. Why not use H2O/EG system directly?

Minor revison:

1.      In Line 443, Fig. S3 should be Figure S4.

Author Response

(The authors gave the same response as above.)
